# Aroma Compounds in Essential Oils: Analyzing Chemical Composition Using Two-Dimensional Gas Chromatography–High Resolution Time-of-Flight Mass Spectrometry Combined with Chemometrics

**DOI:** 10.3390/plants12122362

**Published:** 2023-06-18

**Authors:** Nemanja Koljančić, Olga Vyviurska, Ivan Špánik

**Affiliations:** Institute of Analytical Chemistry, Faculty of Chemical and Food Technology, Slovak University of Technology in Bratislava, Radlinského 9, 812 37 Bratislava, Slovakia; nemanja.koljancic@stuba.sk (N.K.); olga.vyviurska@stuba.sk (O.V.)

**Keywords:** essential oils, comprehensive two-dimensional gas chromatography, tile-based fisher ratio analysis, principal component analysis, stationary phase selection

## Abstract

Analyzing essential oils is a challenging task for chemists because their composition can vary depending on various factors. The separation potential of volatile compounds using enantioselective two-dimensional gas chromatography coupled with high-resolution time-of-flight mass spectrometry (GC×GC–HRTOF-MS) with three different stationary phases in the first dimension was evaluated to classify different types of rose essential oils. The results showed that selecting only ten specific compounds was enough for efficient sample classification instead of the initial 100 compounds. The study also investigated the separation efficiencies of three stationary phases in the first dimension: Chirasil-Dex, MEGA-DEX DET—β, and Rt-βDEXsp. Chirasil-Dex had the largest separation factor and separation space, ranging from 47.35% to 56.38%, while Rt-βDEXsp had the smallest, ranging from 23.36% to 26.21%. MEGA-DEX DET—β and Chirasil-Dex allowed group-type separation based on factors such as polarity, H-bonding ability, and polarizability, whereas group-type separation with Rt-βDEXsp was almost imperceptible. The modulation period was 6 s with Chirasil-Dex and 8 s with the other two set-ups. Overall, the study showed that analyzing essential oils using GC×GC–HRTOF-MS with a specific selection of compounds and stationary phase can be effective in classifying different oil types.

## 1. Introduction

Research of the composition of essential oils is one of the unlimited areas in analytical chemistry. This is primarily due to the complex composition of the raw materials from which the oils are obtained, as well as the plentiful methods of analysis used and the different approaches employed to process the obtained data results [1,2,3,4,5,6,7,8]. This complexity is especially evident in rose essential oils, which are produced by several countries worldwide. The leading producers of rose essential oils include Bulgaria, Turkey, Morocco, Egypt, China, Russia, Ukraine, Moldova, Saudi Arabia, Iran, Afghanistan and India. Rose oil is the most exclusive essential oil, with *Rosa damascena*, *R. alba* and *R. centofilia* being the most important species used for the production of essential oils; rose water, absolute and concrete are then used in the flavor, fragrance, food and pharmaceutical industries [3,9]. Between 3500 and 5000 kg of rose flowers are required for the production of approximately 1 kg of oil. In addition to being often used in folk medicine, rose essential oils are known to possess antimicrobial, analgesic, antispasmodic, muscle relaxant, anti-inflammatory, anticonvulsant, antiviral, antioxidant, anti-HIV and anti-cancer properties [2]. The chemical composition of essential oils strongly depends on the type of rose, climatic conditions, geographical area, harvesting time, storage conditions and production method [2]. Although there are appropriate standards that determine the required chemical composition of rose essential oils (ISO 9842), they are actually only a rough guide due to the complex composition and subsequent production conditions. The quality of the essential oil can be determined based on the composition of the main organic components, alcohols, hydrocarbons, phenols, aldehydes, esters and ketones, as well as certain sulfur compounds [1,2,10]. However, in many studies, a large difference in the concentrations of individual compounds has been observed for the same type of essential oil [4]. It is assumed that the main cause is the application of different extraction methods, which in some cases, leads to a partial or significant loss of the corresponding components, as well as climatic conditions. One example is the study of Antonelli et al., in which citronellol was not found even in traces in some cultivars of *R. damscena*, *R. alba*, *R. centifolia*, *R. gallica* and *R. rugosa*, while in other works, this compound was one of the main constituents of essential oils [1,2,11,12]. For the latter reasons, it is necessary to resort to the adoption of more precise analytical techniques that provide accurate results as unambiguously as possible. As one such method, comprehensive two-dimensional gas chromatography coupled with high-resolution time-of-flight mass spectrometry (GC×GC–HR-TOFMS) might be very efficient. Combining GC×GC with a HR-TOFMS system has been found to be very useful, since a high separation space as well as high acquisition rates (minimum 100 spectra/s) can be attained. Finding statistically significant values in the obtained data that contain tables with hundreds and thousands of compounds is an exhaustive and time-consuming process; however, in the last few years, a large number of commercial software that enable faster and more reliable data processing (GCImage, ChromaTOF, Chromspace, ChromCompare+, GC×GC-Analyzer, StillPeaks GC×GC, ChromSquare, HyperChrom) have been developed. After overcoming the inadequacies of the pixel- and peak table-based F-ratio approach to processing GC×GC chromatograms, Prof. Synovec and his group developed ChromaTOF-Tile software that locates significant spots on chromatograms, determines the nominal mass areas, and then uses the integrals of areas to calculate the F-ratio based on one-way analysis of variance (ANOVA) [13,14]. The size of the tiles is uniform across all chromatograms and is arbitrarily determined by the user. Ideally, the tile dimensions should be such that each tile includes only one peak and possible run-to-run shifting with the amount of within-class sample variance. However, this is very challenging to achieve in practice, mainly when dealing with chromatograms with peak areas that differ by several orders of magnitude. It is assumed that the features in the tiles that have higher F-ratio values are potentially discriminatory and can be further treated and subjected to identification, which significantly shortens the processing time of the obtained data [15,16,17]. This software has recently been successfully used in various non-target analyses, such as metabolomics [18,19,20,21], food and beverage samples [22,23], the analysis of organic compounds in catalysts and organic-based getters [24], as well as target and non-target analyses of environmental samples [25,26].

In this paper, the composition of the volatile organic compounds of five different rose essential oil samples, with three replicates, obtained using the GC×GC–HRTOF-MS method and three different orthogonal combinations of stationary phases, was analyzed. The primary goal was to develop a method for the reliable and rapid identification of discriminative volatile compounds in different samples of essential oils (*R. damscena*, *R. alba*, *R. centifolia*, *R. Gallica* and *Prasophyllum roseum*), with the support of the tile-based Fisher ratio (F-ratio) chemometric approach. Furthermore, the investigation also encompassed an evaluation of the separation efficacy of three enantioselective columns in the first dimension [27,28,29].

## 2. Results and Discussion

The potential of separating the volatile compounds in the samples of the five different types of rose essential oil was demonstrated using GC×GC–HRTOF-MS, with the use of three different stationary phases in the first dimension, namely Chirasil-Dex-permethylated-β-cyclodextrin; MEGA-DEX DET—β—2,3-diethyl-6-tertbutylsilyl-β-cyclodextrin; and Rt-βDEXsp-2,3-dipropyl-6-tertbutylsilyl-β-cyclodextrin, to find the best resolution. Figure 1 shows the chromatogram of the *R. centifolia* sample analyzed on column set-up 1.

In order to identify important characteristics in the raw data and non-target analysis sample set, a tile-based F-ratio analysis method was employed. The analysis reveals the presence of various groups of compounds, such as terpenes, esters, ethers, alcohols, aldehydes, ketones, and phenols, resulting in a total of 3306, 3824, and 4038 features for column set-ups 1, 2, and 3, respectively. Some of these features may represent false positives, redundant hits, siloxanes from the stationary phase, or could be a result of the overlap of two or more features in one tile. Compared to the GC–MS systems used in different papers [30,31,32,33,34,35,36], a significantly larger number of compounds were identified using the GC×GC system, which was allowed by a high separation space, resolution, and the use of HRTOF-MS detectors with a high acquisition rate. Previous research has shown that a large number of compounds cannot be identified using only 1D GC systems due to coelutions primarily arising in the second dimension. Therefore, GC×GC is often necessary for efficient separation, as each identified compound possesses a complex and unique thermodynamic separation mechanism [30,31,32,33,34,35,36]. In addition, due to the focusing effect caused by the modulator, GC×GC provides a much higher *S*/*N* ratio compared to GC methods, providing greater sensitivity and easier eventual quantification. In this study, tile-based F-ratio analysis enabled the relatively easy detection and identification of features; however, ChromaTOF software was also used to confirm the identified compounds. A major advantage of tile-based F-ratio analysis is that specific mass channels can be explored and identified. However, despite the use of efficient software for quickly detecting differences within tiles, the analysis results must meet rigorous requirements, such as a high resolution, sensitivity, and peak capacity, suitable for efficient quantification.

The distinctive aroma of roses and rose oil comes from the complex composition of compounds that diffuse into the air. Previous research has indicated that citronellol, geraniol, and menthone are the most common primary constituents of graveolens essential oils [37,38,39]. In addition to these, citronellyl formate, guaia-6,9-diene, isomenthone, and 10-epi-γ-eudesmol are present in higher concentrations with certain variations depending on the plant’s place of cultivation, with the greatest variations observed in caryophyllene oxide. Menthone, myrcene, δ-cadinene, linalool, geranyl formate, geranyl acetate, geranial, and spathulenol are also important constituents of this essential oil [40,41,42,43,44,45]. *R. gallica* is generally characterized by a higher content of geraniol, citronellol, alkanes, and phenylethyl alcohol [46]. It also contains a similar composition of hydrocarbons in general. Similarly, *R. alba* essential oil has a comparable composition, with an increased concentration of citral, caryophyllene, α-muurolene, and β-cubebene, some aldehydes such as (E)-2-hexenal, (E)-2-heptenal and (E, E)-2,4-heptadienal, and oxygenated sesquiterpenes [1,47,48]. Mahboubi and co-workers found that terpene alcohols, β-citronellol, nerol, geraniol, β-phenyl ethyl benzoate, phenyl ethyl alcohol, nonadecane, methyl eugenol, eugenol, heneicosane, benzyl alcohol, and β-fenchyl alcohol dominate in samples of *R. damascena* essential oil [49]. However, as with most other essential oils, geraniol and citronellol are the main constituents [5,49,50,51,52]. Although hydrocarbons are not highly valued in high-quality rose oils, they still make up a significant portion of the essential oil, alongside compounds that contribute to aroma [53,54,55]. Berechet et al. identified 42 compounds in *R. damascena* essential oil and found that aliphatic hydrocarbons accounted for 85.76%, while monoterpenes only made up 6.54% [56]. *R. damascena Mill*., from southern Iran, has the highest content of nonadecane and heneicosane, while the same variety in Bulgaria exhibits the highest levels of geraniol and citronellol. *R. centofilia* is generally characterized by higher levels of terpene alcohols, phenylethyl alcohol, geraniol, and citronellol, but also high levels of aliphatic hydrocarbons, which often lead to its avoidance in the perfume industry [2,3,9]. This is why *R. alba* and *R. damascena* are considered the most desirable essential oils in the perfume industry [50,56]. According to previous research, *P. roseum* has a very similar composition to rose essential oil, with geraniol, citronellol, geranyl formate, citronellyl formate, linalool, and isomenthone being the most important constituents [40,57]. Depending on the origin, *P. roseum* can vary in its composition; therefore, the essential oil from southwestern Iran also contains caryophyllene, citronellyl acetate, menthone, δ-selinene, δ-cadinene, and β-bourbonene [57,58,59], while the essential oil from southern Africa contains citronellyl butyrate, geranyl tiglate, geranyl, guaia-6,9-diene, and gamma-eudesmol as the main constituents [58]. This study focused only on terpene compounds that contribute to the aroma of essential oils, excluding aliphatic compounds.

In order to confirm the identification of the selected features, accurate mass, and mass spectral match factors, including similarity and probability, were utilized. Probability and similarity were used to determine whether a feature belongs to one compound or multiple compounds and to measure how well the mass spectrum of the feature matches with the database spectrum. According to some of the literature, a similarity above 700 or 800 indicates a match with the database spectrum [34]. Therefore, for all 100 compounds in each set-up, all compounds with a similarity greater than 700 were identified, except for column set-up 3, where *n*-dodecenylsuccinic anhydride and 7α-hydroperoxy manool showed a similarity less than 700. For their confirmation, ChromaTOF software was used. The enantiomeric composition (*EC*) of essential oil samples was calculated in order to determine the percentage ratio of one enantiomeric form and the sum of all enantiomers of a chiral compound, according to Formula (4):(1)$$EC=\frac{A_{i}}{\Sigma A}\times 100\%$$
where $A_{i}$ represents the area of the modulated peaks of the dominant enantiomer, and *A* is the total area of all enantiomers of the same compound. Table 1 presents the enantiomeric composition of the chiral compounds obtained from the analysis conducted on column set-up 1 in the studied samples. The enantiomeric compositions of the samples analyzed on other column set-ups are shown in Appendix A.

Table 2 and Appendix A show the aroma compound composition of essential oils based on the F-ratio for the top 100 compounds according to the F-ratio analysis, belonging to the alcohol, aldehyde, ketone, terpene, ester, and lactone groups. There are significant differences in the composition of essential oils, which are mostly attributed to different climatic conditions and types of roses [44,60,61].

Rose oxide was the only compound separated using all three set-ups. Additionally, citronellol and β-bourbonene were separated using set-ups 1 and 2, while spathulenol and β-eudesmol were separated using column set-ups 2 and 3 (Table 2 and Appendix A). In order to achieve a better resolution, thicker stationary phases or longer nonpolar columns are usually used in the first dimension, while the second column is polar or moderately polar and short. Table 3 presents the values calculated for the total and used separation space, expressed in pixels, as well as the percentage of separation space on the chromatograms across all samples and column set-ups. When employing column set-ups 1, 2, and 3, the greatest separation space was observed using Chirasil-Dex, ranging from 47.35% to 56.38%, while the smallest was observed with the Rt-βDEXsp stationary phase, ranging from 23.36% to 26.21%. It should be noted that a modulation period of 6 s was used with the Chirasil-Dex stationary phase, while an 8 s modulation period was used in the other two column set-ups. Moreover, when using the MEGA-DEX DET—β and Chirasil-Dex stationary phases, group-type separation can be observed (i.e., same polarity, H-bonding ability, polarizability, etc.), while group-type separation using Rt-βDEXsp is almost imperceptible, as the most features were found in this column set-up.

Another parameter that was found to evaluate the separation efficiency when determining enantiomers that usually co-elute during 1D GC analysis is the separation factor (*α*), which indicates whether and how well two compounds are separated. Even when a certain enantioselective stationary phase is used, enantiomers can often co-elute with other non-chiral compounds. [62,63]. One of the solutions is the application of GC×GC analysis, which achieves orthogonality to make the separation mechanisms in both columns independent of each other. The quality of enantiomeric separation was evaluated using the values of *α* according to the following formula:(2)$$\alpha =\frac{k_{2}}{k_{1}}=\frac{t_{R,2}^{\prime \;}}{t_{R,1}^{\prime \;}}$$
where $k_{2}$ and $k_{1}$ represent separation factors, and $t_{R,2}^{\prime \;}$ and $t_{R,1}^{\prime \;}$ represent the reduced retention times of the second and first enantiomers, respectively. Table 1 shows the separation factors for the identified chiral compounds using column set-up 1, while Appendix A display the retention factors for chiral compounds separated using column set-ups 2 and 3, respectively. From the obtained results, it can be seen that most of the chiral compounds were separated using column set-ups 2 and 1, while only four chiral compounds were separated using column set-up 3, which also had the lowest separation factor values.

The PCA results for column set-up 1 (Figure 2a) depict the clear separation of two basic clusters: rose essential oils in one and the *P. roseum* essential oil in the other. Similar results have been achieved using both column set-up 2 and column set-up 3 (Appendix A).

In all analyses, PC1 separated the *P. roseum* essential oils from the rose essential oils, while only in the case of column set-up 3 there was an apparent difference in the PC2 plot between *R. alba*, *R. centofilia*, and *R. gallica* on one side and *R. damascena* on the other. Based on the loading plots for all column set-ups (Figure 2b), it can be observed that almost all compounds influence the sample clustering. Therefore, it could be assumed that similar results could be obtained by selecting only a smaller number of features. HCA further confirmed the PCA results by providing information on the interconnectedness and mutual similarity of rose oils and the clear distinction of *P. roseum* using all three different column set-ups (Figure 3 and Appendix A).

To select the most relevant compounds, Yamamoto and colleagues employed a method that involved choosing only the top 10 compounds with the highest F-ratios across all three column set-ups (Appendix A) [25,64]. The first 10 compounds, according to the F-ratio, that contribute to aroma were identified as phenethyl alcohol, phenylethyl tiglate, and citronellyl formate in all three column set-ups. In column set-ups 1 and 2, geraniol, nerol, and citronellol were identified, while farnesol and epi-γ-eudesmol were confirmed only in column set-ups 2 and 3. In addition to the aforementioned compounds, methyleugenol, 2-phenylethyl acetate, geranial, and geraniol acetate were identified in column set-up 1. β-eudesmol and β-myrcene were identified in column set-up 2, and menthone, geranyl formate, linalool I, linalool II, and linalyl acetate were identified in column set-up 3. PCA plots in Appendix A were constructed for the first 10 features separated by column set-ups 2 and 3, which showed explained variance totals of 87.67% (60.24% for PC1 and 27.43% for PC2) and 98.08% (90.98% for PC1 and 7.10% for PC2), respectively. On the other hand, the Chirasil-Dex column exhibited the highest value of explained variance, at 99.76% (Figure 4a). This was also the only PCA in which discrimination between the rose oil samples and *R. damascena* essential oil samples was clearly visible from the other samples.

On all three PCA plots using the first ten selected features, there is a noticeable distinction between the *P. roseum* sample and the other rose essential oils. In the loading plot (Appendix A) using the Rt-βDEXsp set-up, the *R. gallica* and *R. alba* samples are characterized by a higher presence of farnesol, while phenyl ethyl alcohol is observed in higher concentration in the samples of *R.centifolia* and *R. damascena* cv. “Janina”. The PCA loading plots for the (Appendix A) analysis using the MEGA-DEX DET—β column set-up indicate that the samples of *R. alba* and *R.centifolia* are characterized by β-eudesmol I, the *R. gallica* sample is characterized by farensol and geraniol, the *R. damascena* cv. “Janina” sample is characterized by phenyl-ethyl alcohol and nerol, and the *P. roseum* sample is characterized by a higher presence of phenylethyl tiglate, citronellyl formate, epi-γ-eudesmol, and β-myrcene. Regarding the Chirasil-Dex column, it is evident that the samples of *R. gallica*, *R. alba*, and *R. centifolia* form a separate cluster, in which 2-phenylethyl acetate, methyleugenol, phenethyl alcohol, geraniol, geranial, and nerol affect the separation of sample *R. damascena* cv. “Janina”; meanwhile, phenylethyl tigelate and citronellyl formate are characteristic of the *P. roseum* sample (Figure 4b).

The greatest advantage of using this method is that it does not require additional preliminary data processing, and the results obtained from the PCA analysis are highly consistent with the clustering of samples based on previously used compounds.

To further validate our results, we performed an HCA analysis, which provides more profound information on how the clusters are related (Figure 5 and Appendix A).

## 3. Materials and Methods

### 3.1. Chemicals, Materials and Samples

The essential oils were obtained from the Institute for Roses and Aromatic Plants, Kazanlak, Bulgaria. The plant material of the main oil-bearing species, *Rosa gallica*, *Rosa alba*, *Rosa centifolia* and *Prasophyllum roseum*, was collected from their own fields in 2019, while *Rosa damascena* cv. “Janina” was collected during Scobelevo harvest in 2019. The sample labels of the essential oils are displayed in Table 4.

### 3.2. Instrumentation

The samples were analyzed using the Pegasus GC×GC–HRTOF-MS (LECO Corporation, St. Joseph, MI, USA) system equipped with an Agilent 7890B (Agilent Technologies, Palo Alto, CA, USA) gas chromatograph coupled to a HRTOF-MS (Leco, San Joseph, MO, USA) and ZX-2 non-cryogenic dual-stage thermal loop modulator. Chromatographic separation was performed using following primary column set-ups:

Set-up 1: Chirasil-Dex—permethylated-β-cyclodextrin (25 m × 0.25 mm i.d. × 0.25 µm film thickness, Varian, Walnut Creek, CA, USA).

Set-up 2: MEGA-DEX DET—β—2,3-diethyl-6-tertbutylsilyl-β-cyclodextrin (25 m × 0.25 mm i.d. × 0.25 µm film thickness, MEGA s.n.c. Capillary column laboratory, Legano, Milan, Italy).

Set-up 3: Rt-βDEXsp—2,3-dipropyl-6-tertbutylsilyl-β-cyclodextrin (25 m × 0.25 mm i.d. × 0.25 µm film thickness, Restek Corporation, Bellefonte, PA, USA).

The secondary column was a mid-polar 50% methyl–50% phenyl siloxane-type stationary-phase Rxi-17Sil (0.56 m length, 0.25 mm i.d., 0.25 µm film thickness, Restek Corporation, Bellefonte, PA, USA). The initial temperature of the primary oven was 40 °C (4 min holdup time). It was then gradually raised to 70 °C at 5 °C min^−1^ and then at 1.5 °C min^−1^ to final temperatures of 190 °C, 200 °C and 210 °C (10 min holdup time) for set-ups 1, 2 and 3, respectively. The temperature of the second oven was maintained in a 5 °C offset compared to the first oven temperature program in all 3 set-ups. A modulator was kept at a 15 °C higher temperature compared to the actual oven temperature. The modulation period was set to 8 s with a hot pulse duration of 2.6 s for column set-up 1 and 2, and 6 s with a hot pulse duration of 1.8 s for set-up 3. Helium (99.999% purity) was used as a carrier gas, supplied with a constant flow rate of 1.0 mL min^−1^. A 1μL aliquot was injected using a split mode at a ratio of 1:200 at a constant temperature of 210 °C. The MS transfer line temperature was set to 220 °C. The mass spectra were obtained at 70 eV ionization energy with ion source temperature set to 250 °C. The mass range of 29–450 *m*/*z* was monitored at the signal acquisition rate of 100 spectra/s.

### 3.3. Data Analysis

Since there were five different samples, three replicates were performed within each class for every column set-up. The GC×GC–HRTOF-MS raw data of all samples were imported from LECO’s ChromaTOF software into ChromaTOF Tile (version 1.01.00.0) by converting the data to .smp files. Firstly, the imported files were processed in a way that meant the optimal tile sizes were determined using a built-in calculator tool. The window size for tile first dimension, ^1^D, was 5 modulations, while the tile second dimension, ^2^D, was 10 spectra. A signal-to-noise ratio (*S*/*N*) threshold of 10 times the noise calculated on a per-tile basis for each *m*/*z* was applied in order to ignore the detector signal and exclude *m*/*z* with low *S*/*N* from further processing. In order to avoid false positives, the F-ratio threshold is usually allocated; in this case, it was 20, since it was found that tiles with an F-ratio less than 20 are unreliable [27]. Tile dimensions were chosen to capture the peak retention times in both the first and second dimensions, but also to include short time shifts that would usually occur during analysis. On the other hand, the dimensions of the tiles need be large enough in order to preserve the selectivity of the individual peaks, “features”, and to avoid false positives. A tile that completely covers the entire feature will have a higher F-ratio for each *m*/*z* compared to a tile that contains parts of the features, since in the latter case, the tile will have a higher within-class variation due to phasing. In the case in which two or more features are overlapped in a single tile, the F-ratio for each *m*/*z* enables the easier identification of the overlap and provides additional information for feature identification [13]. It would be ideal to identify one feature in one tile, however parts of a feature can be found in 2 or more tiles and are then called redundant hits. Supervised statistical analysis based on the F-ratio was applied in order to rank the tiles in the tile hitlist according to the decreasing F-ratio value. The F-ratio is actually the class-to-class variation in the detected signal divided by the sum of the within-class variations in the signal. The class-to-class variation is calculated according to Equation (1):(3)$$\sigma_{cl}^{2}=\frac{\Sigma {\left({\bar{x}}_{i}-\bar{x}\right)}^{2}n_{i}}{\left(k-1\right)}$$
where ${\bar{x}}_{i}$ is the mean of the *i*th class, $\bar{x}$ is the overall mean, $n_{i}$ is the number of measurements in the ith class, and *k* is the number of classes. The within-class variation is calculated as shown in Equation (2):(4)$$\sigma_{err}^{2}=\frac{\Sigma \left(\Sigma {\left({\bar{x}}_{ij}-\bar{x}\right)}^{2}\right)-\left(\Sigma {\left({\bar{x}}_{i}-\bar{x}\right)}^{2}n_{i}\right)}{\left(N-k\right)}$$
where ${\bar{x}}_{ij}$ is the *i*th measurement of the *j*th class and *N* is the total number of samples. The F-ratio is calculated based on Equation (3):(5)$$Fisher\;ratio=\frac{\sigma_{cl}^{2}}{\sigma_{err}^{2}}=\frac{class-to-class\;variation}{within-class\;variation}$$

The registered signal within each tile is automatically integrated and assigned several characteristic *m*/*z* using the characteristic F-ratio. In this way, the tile-based method enables the quick comparison of raw data without special preprocessing, just by comparing the same feature that appears in the same tiles across the samples. The selection of appropriate *m*/*z* from the hitlist is based on checking all *m*/*z* via visual inspection and comparison with mass spectra databases, but the most targeted are those in which the area difference between classes is the largest for the feature list [19]. Therefore, features with a similarity less than 600 were listed as “unidentified compounds”. After applying *S*/*N* and the F-ratio threshold, we were left with 3306, 3824 and 4038 features, for column set-ups 1, 2 and 3, respectively. Certainly, in addition to true positives, all the listed features contain redundant peaks and false positives. Therefore, for further analysis, we chose the top 10 and 100 features in each column set-up based on the highest F-ratio, which were characterized by *m*/*z* ions and would contribute to the most effective discrimination using clustering techniques with two statistical tools, namely principal component analysis (PCA) and hierarchical cluster analysis (HCA), using a dataset consisting of a 15 by 100 matrix. Before each analysis, the data were auto-scaled, and the analysis was conducted using MATLAB 2021a software. Initially, the analysis was performed on the complete set of samples and all one hundred compounds. These features were then analyzed and identified using the NIST mainlib and FFNSC version 4.0 library databases. Once all relevant features were identified, the composition of the essential oils obtained using different stationary phases in the first dimension was compared.

Since this study focuses on compounds that are commonly present in chiral forms, three different enantioselective stationary phases were used in the first dimension. To assess the separation performance of all three column set-ups, the percentage of separation space occupied by the sample components and separation factors was utilized. The separation space was calculated as the ratio of the 2D area occupied by all components (from the first to the last elution in both dimensions) to the total 2D chromatogram area, subtracted by the hold-up time. Since the separation mechanism, and consequently the peak locations on the 2D chromatogram, are not solely dependent on the selectivity of the stationary phases but also on other parameters such as temperature, flow rate, and injection mode, it can be said that the separation space represents the degree of separation efficiency of the method, rather than just a measure of the stationary-phase selectivity [28,29].

## 4. Conclusions

The possibility of processing enriched information in complex samples using a simpler software tool than that used in classical 2D chromatogram processing has been demonstrated. Tile-based Fisher ratio analysis enables rapid data processing by automatically identifying statistically significant differences between different sample classes, allowing more time to be spent on significant results rather than manually identifying the chromatogram regions responsible for sample discrimination. However, this approach to data analysis also offers unexplored possibilities for addressing separation and analysis issues in 1D GC, one dimensional liquid chromatography, 1D LC, and 2D LCxLC systems, which could lead to the faster implementation of the obtained results in order to improve sample quality and create new software systems for detecting deeply hidden chromatographic information. The PCA plots revealed a significant difference between the *P. roseum* sample and other rose essential oils. Chirasil-Dex had the largest separation space and the most suitable separation factors, with a 6 s modulation period; meanwhile, Rt-βDEXsp had the smallest separation space. The MEGA-DEX DET—β and Chirasil-Dex stationary phases exhibited group-type separation possibilities.

## Figures and Tables

**Figure 1 plants-12-02362-f001:**
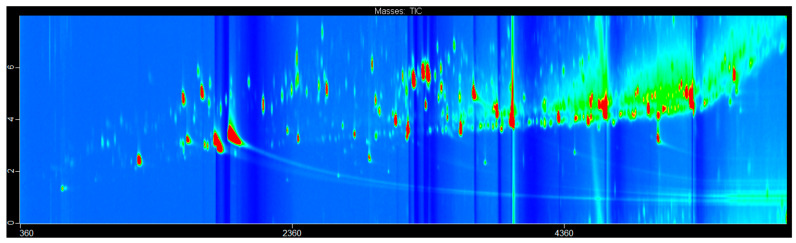
Representative total ion current (TIC) GC×GC chromatograms of *R. centifolia* essential oil obtained using the Chirasil-Dex column set-up.

**Figure 2 plants-12-02362-f002:**
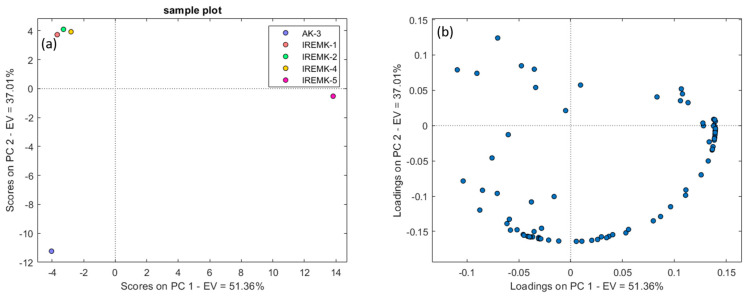
PCA results—score (**a**) and loading (**b**) plots for 100 identified features, with an indication of the discrimination between different essential oils on column set-up 1.

**Figure 3 plants-12-02362-f003:**
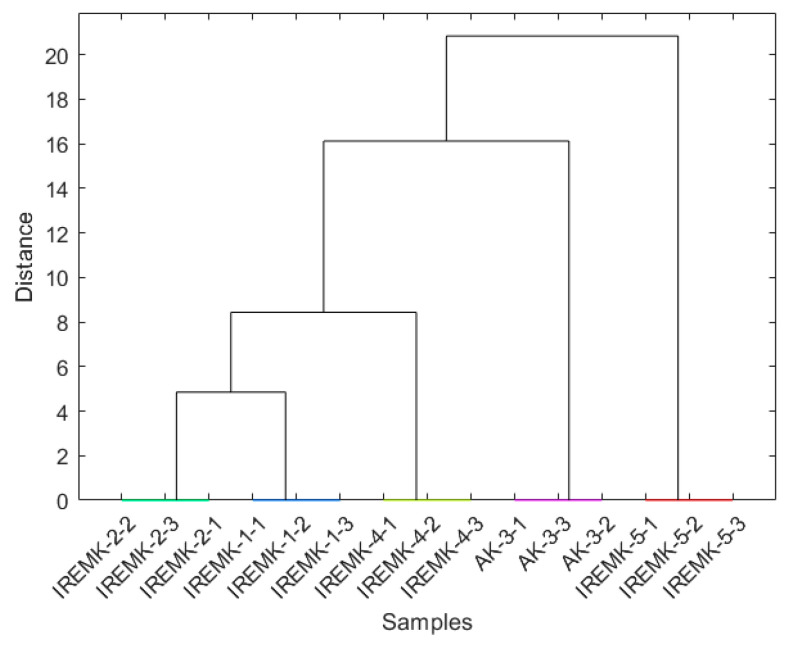
Dendrogram obtained for the 100-feature set obtained on column set-up 1.

**Figure 4 plants-12-02362-f004:**
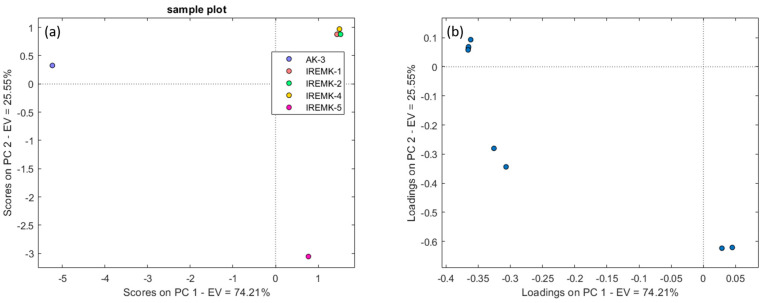
PCA results—score (**a**) and loading (**b**) plots for 10 identified features with indication of discrimination between different essential oils on column set-up 1.

**Figure 5 plants-12-02362-f005:**
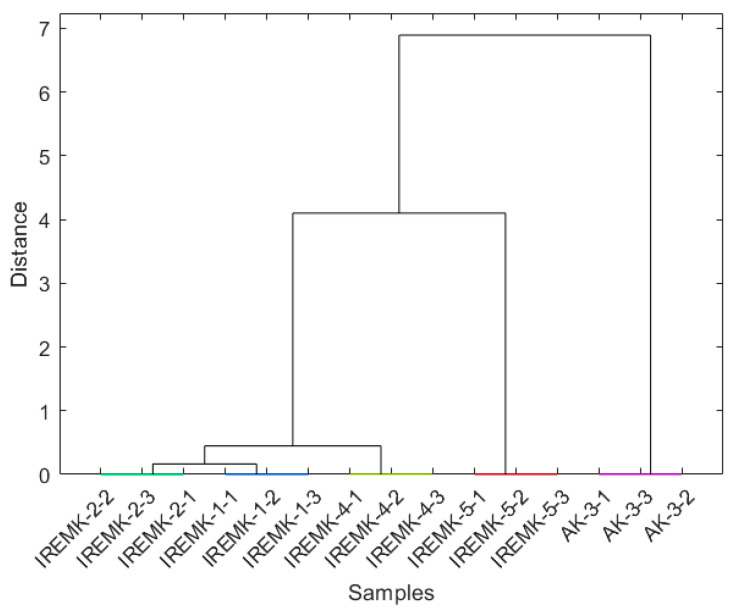
Dendrogram obtained for the 10-feature set obtained using column set-up 1.

**Table 1 plants-12-02362-t001:** Enantiomeric composition and separation factors of selected chiral terpenes in studied essential oils obtained using column set-up 1.

Feature No.	Compound	*EC*, %					*k*
		*R. damascena* cv. “Janina”	*R. gallica*	*R. alba*	*R.centifolia*	*P. roseum*	
1	Citronellol I	89.27	49.57	14.92	43.99	91.13	1.00
2	Citronellol II	10.73	50.43	85.08	56.01	8.87	
3	Citronellyl propionate I	46.33	35.56	10.60	20.51	81.18	1.87
4	Citronellyl propionate II	53.67	64.44	89.40	79.49	18.82	
5	Epicubenol I	5.22	8.31	33.00	2.20	34.06	1.04
6	Epicubenol II	94.78	91.69	67.00	97.80	65.94	
7	Farnesol I	96.67	89.37	87.86	90.52	43.43	1.02
8	Farnesol II	3.33	10.63	12.14	9.48	56.57	
9	Rose oxide I	67.15	67.98	68.65	68.72	69.15	1.10
10	Rose oxide II	32.85	32.02	31.35	31.28	30.85	
11	α-epi-Muurolol I	56.79	86.29	48.92	64.98	55.00	1.03
12	α-epi-Muurolol II	43.21	13.71	51.08	35.02	45.00	
15	α-Guaiene III	38.34	42.92	33.42	36.95	38.02	1.00 1.16
13	α-Guaiene I	31.84	32.92	34.18	28.92	46.75	
14	α-Guaiene II	29.82	24.16	32.40	34.14	15.23	
16	β-Bourbonene I	56.95	1.87	3.28	8.73	9.47	1.01 1.45
17	β-Bourbonene II	42.84	94.86	75.76	86,45	74.00	
18	β-Bourbonene III	0.21	3.27	20.97	4.82	16.53	

*EC*—Enantiomeric composition.

**Table 2 plants-12-02362-t002:** The 100 features identified using the Chirasil-Dex stationary phase in ^1^D as a true positive via tile-based F-ratio analysis, along with retention information.

Feature No.	Name	Similarity	Probability	CAS	Quant Mass	F-Ratio	Mean ^1^t_R_ (Sec)	Mean ^2^t_R_ (Sec)	%RSD ^1^t_R_	%RSD ^2^t_R_
1	Geraniol	889	5831	106-25-2	69	3.30 × 10^15^	1856	3.58	0	3.85 × 10^−14^
2	Nerol	922	8214	106-25-2	69	1.95 × 10^15^	1720	3.45	0	2.66 × 10^−14^
3	Phenethyl alcohol	943	9536	60-12-8	91	1.43 × 10^15^	1560	3.37	0	2.73 × 10^−14^
4	Citronellol I	949	5840	106-22-9	69	1.08 × 10^15^	1752	3.13	0	1.47 × 10^−14^
5	Phenylethyl tigelate	945	9688	55719-85-2	104	1.02 × 10^15^	2952	0.44	0	3.92 × 10^−14^
6	Citronellyl formate	954	6352	105-85-1	69	4.17 × 10^14^	1624	4.20	0	2.19 × 10^−14^
7	Methyleugenol	922	8332	93-15-2	178	3.30 × 10^14^	2200	0.38	0	1.51 × 10^−14^
8	2-Phenylethyl acetate	961	8704	103-45-7	104	2.94 × 10^14^	1656	5.98	0	3.07 × 10^−14^
9	Geranial	892	7768	141-27-5	69	2.54 × 10^14^	1664	5.28	0	0
10	Geraniol acetate	878	2691	25905-14-0	69	2.29 × 10^14^	2032	5.61	0	0
11	Eugenol	937	6353	97-53-0	164	2.25 × 10^14^	2248	5.57	0	3.30 × 10^−14^
12	γ-Eudesmol	921	5617	1209-71-8	189	1.82 × 10^14^	3152	5.95	0	3.09 × 10^−14^
13	Menthone	944	5109	10458-14-7	112	1.21 × 10^14^	1216	4.33	0	2.12 × 10^−14^
14	Rose oxide I	881	9509	16409-43-1	139	1.10 × 10^14^	960	3.24	0	4.26 × 10^−14^
15	Isomenthone	943	5481	491-07-6	112	9.76 × 10^13^	1168	4.05	0	2.27 × 10^−14^
16	β-Myrcene	896	4554	123-35-3	69	8.61 × 10^13^	1720	4.89	0	0
17	α-Humulene	926	7531	6753-98-6	93	8.21 × 10^13^	2272	5.01	0	0
18	Linalool	941	8734	78-70-6	71	7.56 × 10^13^	1216	2.55	0	0
19	Linalyl acetate	899	3336	115-95-7	93	6.56 × 10^13^	1448	4.30	0	2.14 × 10^−14^
20	Geranyl tiglate	858	4553	7785-33-3	93	5.20 × 10^13^	3296	7.22	0	0
21	β-Bourbonene II	851	5710	119903-95-6	81	4.76 × 10^13^	1872	4.72	0	0
22	Citronellol acetate	939	3760	150-84-5	81	4.64 × 10^13^	1840	5.00	0	0
23	trans Calamenene	814	9135	73209-42-4	159	4.55 × 10^13^	2568	6.27	0	2.93 × 10^−14^
24	Farnesol I	909	2231	3790-71-4	69	4.08 × 10^13^	3688	5.16	0	3.56 × 10^−14^
25	Citronellyl tiglate	925	5599	24717-85-9	81	3.57 × 10^13^	3128	6.25	0	0
26	α-Terpineol	932	7680	98-55-5	59	2.75 × 10^13^	1672	3.19	0	1.44 × 10^−14^
27	Lavanduol acetate	935	5589	25905-14-0	69	2.45 × 10^13^	1584	4.74	0	0
28	Rose oxide II	901	9528	16409-43-1	139	2.39 × 10^13^	1048	3.34	0	2.75 × 10^−14^
29	Valerianol	947	4303	20489-45-6	161	2.26 × 10^13^	3336	5.82	0	3.16 × 10^−14^
30	α-Pinene	941	6777	80-56-8	93	2.16 × 10^13^	568	5.83	0	1.58 × 10^−14^
31	Neryl propionate	893	3355	105-91-9	69	1.97 × 10^13^	2392	5.86	0	0
32	Geranyl isobutyrate	855	1869	2345-26-8	69	1.95 × 10^13^	2744	5.90	0	1.56 × 10^−14^
33	Germacrene D	938	5363	23986-74-5	161	1.86 × 10^13^	2360	5.37	0	0.0 × 1000
34	β-Phenylethyl butyrate	947	6918	103-52-6	104	1.62 × 10^13^	2168	6.31	0	2.91 × 10^−14^
35	Linalol acetate	777	765	115-95-7	69	1.59 × 10^13^	1912	5.48	0	3.36 × 10^−14^
36	α-Guaiene III	940	5599	3691-12-1	105	1.56 × 10^13^	2192	4.51	0	2.04 × 10^−14^
37	Citronellyl butyrate	966	6244	141-16-2	81	1.44 × 10^13^	2584	5.30	0	1.73 × 10^−14^
38	Caryophyllene	966	5900	13877-93-5	93	1.36 × 10^13^	2128	4.68	0	1.96 × 10^−14^
39	Dihydromyrcenol	899	8797	18479-58-8	59	1.30 × 10^13^	2784	2.93	0	0
40	Terpinen-4-ol	902	6816	562-74-3	93	1.23 × 10^13^	1520	3.20	0	1.44 × 10^−14^
41	Phenethyl propionate	944	7758	122-70-3	104	1.17 × 10^13^	2008	6.54	0	4.22 × 10^−14^
42	α-Copaene	932	5272	138874-68-7	105	1.05 × 10^13^	1824	4.50	0	0
43	α-Guaiene II	883	1031	3691-12-1	107	9.42 × 10^12^	2544	4.94	0	1.86 × 10^−14^
44	γ-Muurolene	942	5736	39029-41-9	161	8.49 × 10^12^	2552	5.33	0	1.72 × 10^−14^
45	β-Citral	910	6118	106-26-3	41	8.36 × 10^12^	1544	5.01	0	0
46	β-Famesene	857	1624	18794-84-8	69	7.95 × 10^12^	3592	4.90	0	0
47	epi-γ-Eudesmol	919	5350	1209-71-8	91	7.56 × 10^12^	3136	5.84	0	3.15 × 10^−14^
48	Ledene	910	2766	21747-46-6	105	7.49 × 10^12^	2488	4.91	0	3.74 × 10^−14^
49	α-Eudesmol	906	5903	473-16-5	59	6.94 × 10^12^	3320	5.91	0	3.11 × 10^−14^
50	α-epi-Muurolol II	937	3695	19912-62-0	95	6.19 × 10^12^	3448	5.28	0	0
51	α-epi-Cadinol	916	4366	5937-11-1	161	6.14 × 10^12^	3432	4.86	0	0
52	Epicubenol I	923	5221	19912-67-5	119	4.73 × 10^12^	3064	5.79	0	3.18 × 10^−14^
53	Citronellyl propionate I	939	5397	141-14-0	69	4.61 × 10^12^	2232	5.06	0	1.82 × 10^−14^
54	Lavandulyl isobutyrate	879	1592	51117-20-5	93	4.47 × 10^12^	2536	5.54	0	3.32 × 10^−14^
55	Citronellol II	913	4838	1117-61-9	67	4.37 × 10^12^	1760	4.36	0	0
56	*n*-Hexanol	951	8404	111-27-3	56	4.36 × 10^12^	672	1.36	0	3.38 × 10^−14^
57	β-Bourbonene I	923	9138	119903-95-6	80	4.32 × 10^12^	1848	4.76	0	0
58	Citronellyl isohexanoate	909	3155	71662-18-5	81	4.27 × 10^12^	3216	5.29	0	1.74 × 10^−14^
59	Methyl geraniate	890	8608	2349-14-6	69	4.06 × 10^12^	1704	5.73	0	3.21 × 10^−14^
60	δ-Cadinene	953	6110	16729-01-4	81	3.97 × 10^12^	2600	5.28	0	0
61	β-Pinene	943	6657	127-91-3	93	3.59 × 10^12^	672	6.14	0	0
62	Farnesal	860	6270	502-67-0	69	3.32 × 10^12^	3552	6.84	0	4.03 × 10^−14^
63	Geranyl propionate	834	2474	105-90-8	69	3.25 × 10^12^	2896	5.72	0	0
64	para-Cymene	884	4466	99-87-6	119	3.20 × 10^12^	712	2.53	0	1.82 × 10^−14^
65	Phytol acetate	865	1716	10236-16-5	95	3.10 × 10^12^	5152	6.03	0	0
66	Neryl hexanoate	880	1283	68310-59-8	69	2.98 × 10^12^	3376	5.98	0	3.07 × 10^−14^
67	α-Calacorene	940	8993	21391-99-1	157	2.85 × 10^12^	2576	7.38	0	2.49 × 10^−14^
68	Spathulenol	944	7254	72203-24-8	91	2.68 × 10^12^	3112	5.17	0	1.78 × 10^−14^
69	β-Bourbonene III	873	7103	5208-59-3	81	2.61 × 10^12^	2672	5.69	0	3.23 × 10^−14^
70	Farnesol II	786	1101	3790-71-4	69	2.11 × 10^12^	3760	4.65	0	0
71	β-Dihydroagarofuran	957	5868	5956-09-2	137	2.01 × 10^12^	2440	5.89	0	0
72	β-Farnesene	912	3136	18794-84-8	69	1.97 × 10^12^	2240	4.70	0	1.96 × 10^−14^
73	Nerolidol	945	4521	40716-66-3	69	1.86 × 10^12^	3000	4.33	0	2.12 × 10^−14^
74	α-Farnesene	943	6646	502-61-4	93	1.72 × 10^12^	2464	5.08	0	1.81 × 10^−14^
75	Geranic acid	956	7882	4698-08-2	69	1.64 × 10^12^	2928	2.52	0	1.82 × 10^−14^
76	Aromandendrene	920	2321	489-39-4	91	1.60 × 10^12^	2312	4.91	0	3.74 × 10^−14^
77	Myrcene	925	7292	123-35-3	41	1.53 × 10^12^	616	1.48	0	1.55 × 10^−14^
78	10-epi-α-Eudesmol	874	4859	473-16-5	59	1.39 × 10^12^	3200	5.09	0	3.61 × 10^−14^
79	2-Phenylethyl dodecanoate	930	6162	6309-54-2	104	1.38 × 10^12^	5312	0.24	0	4.79 × 10^−14^
80	Elemol	916	5631	639-99-6	93	1.32 × 10^12^	2968	4.76	0	0
81	α-Copaen-11-ol	868	8105	41370-56-3	59	1.26 × 10^12^	3448	5.98	0	3.07 × 10^−14^
82	Linalyl butyrate	865	1853	78-36-4	69	1.26 × 10^12^	3520	6.10	0	1.51 × 10^−14^
83	Aciphyllene	941	2394	87745-31-1	105	1.11 × 10^12^	2424	5.26	0	0
84	Epicubenol II	909	5249	19912-67-5	161	1.10 × 10^12^	3184	5.75	0	0
85	*n*-Hexadecanoic acid	908	7998	57-10-3	73	1.04 × 10^12^	5048	3.25	0	0
86	Phenylethyl octanoate	880	6941	5457-70-5	104	9.90 × 10^11^	3944	7.35	0	2.50 × 10^−14^
87	β-Copaene	941	4437	147515-11-5	161	8.78 × 10^11^	2080	4.95	0	1.86 × 10^−14^
88	α-epi-Muurolol I	950	6183	19912-62-0	105	8.12 × 10^11^	3352	5.36	0	0
89	β-Elemene	927	2299	33880-83-0	93	8.12 × 10^11^	2480	5.49	0	1.67 × 10^−14^
90	Benzaldehyde	942	9578	100-52-7	105	8.12 × 10^11^	712	2.99	0	3.07 × 10^−14^
91	Ledol	901	2809	577-27-5	105	7.89 × 10^11^	3080	5.51	0	1.67 × 10^−14^
92	Norbourbonane	904	8790	13844-03-6	81	7.62 × 10^11^	2768	7.56	0	2.43 × 10^−14^
93	Citronellyl propionate II	886	3313	141-14-0	81	7.56 × 10^11^	4080	5.65	0	1.63 × 10^−14^
94	β-Selinene	931	1656	17066-67-0	93	7.24 × 10^11^	2384	5.37	0	0
95	α-Guaiene I	783	1722	3691-12-1	105	7.05 × 10^11^	2200	4.86	0	0
96	Citronellyl caprate	875	999	72934-06-6	95	6.58 × 10^11^	4752	5.95	0	3.09 × 10^−14^
97	Benzyl tiglate	905	9511	37526-88-8	91	6.58 × 10^11^	2584	0.36	0	1.60 × 10^−14^
98	Nonanal	913	6842	124-19-6	57	6.26 × 10^11^	1000	3.12	0	1.47 × 10^−14^
99	α-Muurolene	906	2115	10208-80-7	105	6.17 × 10^11^	2312	5.46	0	3.37 × 10^−14^
100	Linalool oxide	925	7468	5989-33-3	59	5.94 × 10^11^	1096	2.37	0	0

**Table 3 plants-12-02362-t003:** Separation space characteristics based on pixel counts across all column set-ups.

Sample	Column Set-Up 1			Column Set-Up 2			Column Set-Up 3		
	Area Total (Pixel)	Area Used (Pixel)	Separation Space (%)	Area Total (Pixel)	Area Used (Pixel)	Separation Space (%)	Area Total (Pixel)	Area Used (Pixel)	Separation Space (%)
*R. damascena cv.* “Janina”	63.42	35.76	56.38	63.70	31.70	49.77	63.34	14.90	23.52
*R. gallica*	63.22	29.01	45.88	63.34	30.62	48.35	63.35	16.50	26.04
*R. alba*	63.95	30.28	47.35	63.50	25.39	39.98	63.34	14.80	23.36
*R.centifolia*	63.54	31.20	49.11	63.58	28.99	45.60	63.34	16.60	26.21
*P. roseum*	63.34	32.50	51.31	63.34	24.33	38.41	63.34	15.87	25.06

**Table 4 plants-12-02362-t004:** Essential oil samples and their corresponding codes.

Essential Oil	Sample Code
*Rosa damascena* cv. “Janina”	AK3
*Rosa gallica*	IREMK1
*Rosa alba*	IREMK2
*Rosa centifolia*	IREMK4
*Prasophyllum roseum*	IREMK5

## Data Availability

The data presented in this study are available upon request from the corresponding author.

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
