# Peer review of "Aroma Compounds in Essential Oils: Analyzing Chemical Composition Using Two-Dimensional Gas Chromatography–High Resolution Time-of-Flight Mass Spectrometry Combined with Chemometrics"

_plants, 2023, doi:10.3390/plants12122362_

Round 1

Reviewer 1 Report

Congratulations for the well-written manuscript. Suggestion to Editor to accept the paper. 

Author Response

Thank you very much for your opinion.

Reviewer 2 Report

Koljančić and Špánik presented their original research article analyzing essential oils from 5 different plant species. Moreover, the study was extended to a comparison of three different stationary phases with the support of a chemometric approach. Overall, the study seems planned and well done, and only some minor suggestions were provided below.

In my opinion, the proposed title would be more suitable for a review paper than a research article. I suggest mentioning that the paper deals with analysing rose essential oils.

- line 89 - "gallica" should be written in lowercase

- Do you know what method was used to obtain essential oils from the tested plants?

- some Greek letters are missing from the text, e.g. lines 107 or 112

- Figure 1. - italics needed

- Figures 2b and 4b - compound names overlap, making it look not good.

- The reference list should be improved according to the Instructions for authors.

Reviewer 3 Report

The present manuscript describes a method to identify with few compounds the species used in the essential oil analyzed as it is able to descriminate with out problem, despite the high number of compounds that could be detected in one analysis.

The document is gloablly well written but some sentences should be reviewed.

The introduction is complete presents clearly the obejectives of the work.

The material and method is quite complete but there is a part which is reported in the results section that should be moved to this section.

The results are clearly presented and discussed, the only thing that the study do not discuss is how this method will work face to the variability of composition obtained after collecting the essential oil in different places or years, or face to fiddrent extraction techniques which may alterate the presence of some compounds. This should be implemented.

The conclusion is fine.

More detailed comments in the attached file

There are some sentences where the english is difficult to understand

Round 2

Reviewer 3 Report

The authors have provided the corrections requested. Now the document is ready to be published.

The material and method section which had the major problems has been corrected.